# Geriatric Nutritional Risk Index at Hospital Admission or Discharge in Patients with Acute Decompensated Heart Failure

**DOI:** 10.3390/jcm12051891

**Published:** 2023-02-27

**Authors:** Masafumi Ono, Atsushi Mizuno, Shun Kohsaka, Yasuyuki Shiraishi, Takashi Kohno, Yuji Nagatomo, Ayumi Goda, Shintaro Nakano, Nobuyuki Komiyama, Tsutomu Yoshikawa

**Affiliations:** 1Department of Cardiovascular Medicine, St. Luke’s International Hospital, Tokyo 104-8560, Japan; 2Department of Cardiology, Keio University School of Medicine, Tokyo 108-8345, Japan; 3Department of Cardiovascular Medicine, Kyorin University Faculty of Medicine, Tokyo 181-8611, Japan; 4Department of Cardiology, National Defense Medical College, Tokorozawa 359-8513, Japan; 5Department of Cardiology, International Medical Center, Saitama Medical University, Saitama 350-1298, Japan; 6Department of Cardiology, Sakakibara Heart Institute, Tokyo 183-0003, Japan

**Keywords:** Geriatric Nutritional Risk Index, acute decompensated heart failure, nutrition, prognosis

## Abstract

Geriatric Nutritional Risk Index (GNRI) is known both as a reliable indicator of nutritional status and a predictor of long-term survival among patients with acute decompensated heart failure (ADHF). However, the optimal timing to evaluate GNRI during hospitalization remains unclear. In the present study, we retrospectively analyzed patients hospitalized with ADHF in the West Tokyo Heart Failure (WET-HF) registry. GNRI was assessed at hospital admission (a-GNRI) and discharge (d-GNRI). Out of 1474 patients included in the present study, 568 (40.1%) and 796 (57.2%) patients had lower GNRI (<92) at hospital admission and discharge, respectively. After the follow-up (median 616 days), 290 patients died. The multivariable analysis showed that all-cause mortality was independently associated with d-GNRI (per 1 unit decrease, adjusted hazard ratio [aHR]: 1.06, 95% confidence interval [CI]: 1.04–1.09, *p* < 0.001), but not with a-GNRI (aHR: 0.99, 95% CI: 0.97–1.01, *p* = 0.341). The predictability of GNRI for long-term survival was more pronounced when evaluated at hospital discharge than at hospital admission (area under the curve 0.699 vs. 0.629, DeLong’s test *p* < 0.001). Our study suggested that GNRI should be evaluated at hospital discharge, regardless of the assessment at hospital admission, to predict the long-term prognosis for patients hospitalized with ADHF.

## 1. Introduction

Acute decompensated heart failure (ADHF) has a strong relationship with malnutrition [1]; ADHF can induce anorexia, malabsorption secondary to intestinal edema, high energy demand, and cytokine-induced hypercatabolism, which cause severe nutritional deterioration, while malnutrition could also be a major cause of refractory HF. The combination of ADHF and malnutrition synergistically deteriorate the patient’s systemic condition, leading to further worse prognosis [1]. 

Notwithstanding the importance of nutritional assessment for ADHF patients, there is a lack of consensus regarding the methodology for the assessment. Some objective scores have been advocated for the purpose of nutritional evaluation among patients with HF. Geriatric Nutritional Risk Index (GNRI) is one of the objective scores to assess a patient’s nutritional condition. GNRI consists of body mass index (BMI) and albumin, both of which are commonly measured in dairy clinical practice, and therefore, GNRI could be easily calculated and applied even in a busy clinical setting [2]. Previous studies revealed that lower GNRI is an independent predictor of short- and long-term mortality in hospitalized patients with ADHF, when evaluated either at hospital admission or discharge [3,4,5,6,7]. However, the nutritional status of ADHF patients could dramatically change, especially in the acute phase, due to an inflammatory response, energy intake/expenditure balance, and/or underlying disease condition [8,9,10]. Thus far, the optimal timing to assess GNRI for predicting the long-term survival of ADHF patients has not been investigated [11].

The present study aimed to investigate the effects of GNRI assessed at the time of hospital admission and discharge on the long-term survival among hospitalized patients presenting with ADHF.

## 2. Materials and Methods

### 2.1. Study Population

The present study is a sub-study of the West Tokyo Heart Failure Registry (WET-HF Registry). Details of the WET-HF Registry have been described previously and well-validated by prognostic models for patients hospitalized with heart failure [12,13,14,15,16,17]. Briefly, this database is an ongoing, multicenter, prospective cohort, registry study designed to collect data pertaining to the clinical backgrounds and outcomes of patients who were hospitalized with the clinical diagnosis of ADHF, according to the Framingham criteria [18]. Prior to the launch of the registry, the objective and detailed study design were provided for clinical trial registration to the University Hospital Medical Information Network of Japan (UMIN000001171). The study was conducted at six study centers, including three university hospitals and three tertiary referral hospitals. To guarantee the quality of the acquired data, baseline data and outcome were collected from the medical record and/or were obtained by querying treating physicians by dedicated clinical research coordinators if necessary. On-site monitoring was performed by the investigators (Y.S. and S.K.) at least once a year to ensure the proper registration of each patient. Patients who declined to participate in the study or presented with acute coronary syndrome were excluded from the registration. Informed consent was obtained from each subject before enrollment in the study. The study protocol was approved by the institutional review boards at each site, and research was conducted in accordance with the Declaration of Helsinki.

### 2.2. Data Collection and Endpoint

We collected conventional clinical variables, including age, sex, etiology of HF, medical history, vital status at admission and discharge, laboratory data (at admission, during hospitalization, and at discharge), and medications (at admission and discharge). Patients who did not have available data of serum albumin and BMI at hospital admission and/or discharge and at the follow-up vital status were excluded from the current study.

The primary endpoint of the present study was all-cause mortality at the maximum follow-up period from the discharge of the index hospitalization (defined as day 0), which is the most robust clinical endpoint without the necessity of adjudication.

### 2.3. GNRI

We calculated patients’ GNRI at both admission (a-GNRI: GNRI at hospital admission) and discharge (d-GNRI: GNRI at hospital discharge) by the following formula [2]: GNRI = [14.89 × serum albumin (g/dL)] + [41.7 × (body weight/ideal body weight)] = [14.89 × serum albumin (g/dL)] + [41.7 × (BMI/22)]

When a patient’s body weight exceeded the ideal body weight, “body weight/ideal body weight” was set to 1, according to the original criteria [2]. Patients were classified as lower GNRI (<92) with moderate or severe nutritional risk, or higher GNRI (≥92) with low or mild nutritional risk, according to the threshold of 92 [2]. 

### 2.4. Statistical Analysis

Continuous variables were reported as median with interquartile range (IQR) and were compared using the Kruskal–Wallis H test. Categorical variables were presented as counts and percentage and were compared using the chi-square test or Fisher’s exact test as appropriate. 

A scatter plot between a-GNRI and d-GNRI was drawn, depicting the linear regression line with the 95% confidence interval (CI), and the Pearson correlation was used to quantify the relation between a-GNRI and d-GNRI. To visualize the impacts of BMI and albumin on d-GNRI, as well as changes in GNRI from hospital admission to discharge (∆GNRI), we made heatmaps in which d-GNRI and ∆GNRI were color-coded for each observed pair of BMI-albumin at hospital admission and pair of ∆BMI-∆albumin during hospitalization, respectively.

The Kaplan–Meier method was used to estimate the cumulative rates of events, and the log-rank test was performed to examine the differences between groups. The cumulative incidence of all-cause death during the follow-up period was compared between the lower GNRI (<92) and higher GNRI (≥92) groups at both timepoints (hospital admission and discharge) using unadjusted and adjusted Cox proportional hazard models to calculate unadjusted and adjusted HRs with 95% CIs, respectively. Adjusted baseline variables included age (years), sex, hypertension, dyslipidemia, diabetes mellitus, renal failure requiring hemodialysis, current smoking, chronic pulmonary occlusive disease (COPD), history of HF admission, New York Heart Association Functional Classification (NYHA class) 3 or 4, laboratory data (serum hemoglobin, Na, blood urea nitrogen [BUN], and serum creatinine) at discharge, left ventricular ejection fraction (LVEF), use of beta-blockers, use of mineralocorticoid receptor antagonists (MRA), use of statins, use of either angiotensin-converting-enzyme inhibitors (ACEI) or angiotensin II receptor blockers (ARB) at discharge, length of index hospitalization (days), and a-GNRI (for d-GNRI) or d-GNRI (for a-GNRI) that had been selected based on prior knowledge of the association of these covariables with the outcomes [19].

The association between a-GNRI or d-GNRI and the primary endpoint was also assessed as a continuous variable, depicting restricted cubic spline curves derived from the unadjusted proportional hazards models. In order to determine the best variable for predicting the clinical outcome, the area under the receiver-operating characteristic (ROC) curves and areas under the curves (AUC) were estimated for several variables, including a-GNRI, d-GNRI, ∆GNRI, BMI at discharge, ∆BMI, albumin at discharge, and ∆albumin for the primary endpoint, and were compared by using the DeLong method [20].

A two-sided *p* value of less than 0.05 was considered to indicate statistical significance. All analyses were performed in SPSS Statistics, version 29 (IBM Corp., Armonk, NY, USA) and R software version 4.0.1 (R Foundation for Statistical Computing, Vienna, Austria).

## 3. Results

From 2006 to 2017, 4000 patients were enrolled in the WET-HF Registry. Out of those patients, 2410 patients who did not have available GNRI data and 116 patients without any follow-up data were excluded in this study. Finally, a total number of 1474 patients (median age, 76 years [IQR: 65–83]; 58.5% male) were included in the present study (Figure 1). 

### 3.1. Distribution of a-GNRI and d-GNRI

Figure 2 shows the distribution of a-GNRI and d-GNRI, with indication of whether GNRI increased or decreased during the hospitalization for each individual patient. The median GNRI at hospital admission and discharge were 93.8 (IQR: 88.4–99.1) and 90.8 (IQR: 84.6–96.8), respectively. At hospital admission, 906 (61.5%) patients had a higher GNRI (≥92), and 568 (38.5%) patients had a lower GNRI (<92), whereas at hospital discharge, 678 (46.0%) patients had a higher GNRI, and 796 (54.0%) patients had a lower GNRI. Although there was a strong correlation between a-GNRI and d-GNRI with the Pearson correlation coefficient of 0.7 (*p* < 0.001), the majority of the patients showed decreased GNRI from hospital admission to discharge (Figure 2). The median value of the changes in GNRI during hospitalization was −2.95 (IQR: −7.30 to +1.49). As components of GNRI, the mean values of the changes in serum albumin and BMI during hospitalization were −0.1 (IQR: −0.4 to +0.2) and −1.53 (IQR: −2.61 to −0.79), respectively.

Figure 3A,B shows the relationship among BMI, serum albumin, and GNRI. When predicting d-GNRI according to BMI and albumin values at the time of hospital admission (Figure 3A), most of patients who had serum albumin level >3.0 mg/dL at admission had low or mild nutritional risk at discharge (i.e., d-GNRI ≥ 92), unless the patient’s BMI did not exceed 22 at hospital admission. Moreover, the changes in GNRI during the hospitalization mainly depended on the changes in albumin, rather than those in BMI (Figure 3B).

### 3.2. Other Patient Characteristics

Baseline patient characteristics in comparison between patients with lower GNRI and those with higher GNRI, either at hospital admission or discharge, are shown in Table 1. Patients with lower GNRI were older, more often female, and had a higher NYHA classification, higher prevalence of hypertension and valvular disease, and lower prescription rates of ACEI or ARB, MRA, beta-blockers, and statins than those with higher GNRI. Patients with lower GNRI had lower hemoglobin and Na, as well as higher BUN, C-reactive protein, and brain natriuretic peptide (BNP) or NT-proBNP than those with higher GNRI both at admission and discharge. At discharge, compared to patients with higher GNRI, those with lower GNRI had a higher prevalence of hemodialysis, which was not observed at admission, where 56.6% of hemodialysis patients (30/53) were classified as low or mild nutritional risk on admission, whereas 17.0% (9/53) were classified at discharge. Patients with lower d-GNRI had a higher prevalence of valvular disease and a longer length of hospital stay. Importantly, patients with lower d-GNRI had significantly higher LVEF than those with higher d-GNRI.

### 3.3. Clinical Outcomes According to a-GNRI or d-GNRI

During a median follow-up period of 616 days (IQR: 271 to 925 days), 290 of the study patients died. The Kaplan–Meier curves for all-cause death up to 4 years are shown in Figure 4. At the maximum follow-up period, a-GNRI < 92 and d-GNRI < 92 groups had a significantly higher all-cause mortality risk than a-GNRI ≥ 92 and d-GNRI ≥ 92 groups, respectively (both Log-rank *p* < 0.001).

The unadjusted and adjusted hazard ratios with those 95% Cis, according to a-GNRI or d-GNRI, are shown in Table 2. The crude risk of all-cause death was lower in patients with higher a-GNRI, both as a categorical (<92 vs. ≥92, unadjusted HR: 2.09, 95% CI: 1.66–2.63, *p* < 0.001) and a continuous variable (per 1 unit decrease, unadjusted HR: 1.06, 95% CI: 1.04–1.09, *p* < 0.001). Similarly, patients who had higher d-GNRI showed a lower risk of all-cause death (<92 vs. ≥92, unadjusted HR: 3.82, 95% CI: 2.89–5.05, *p* < 0.001, per 1 unit decrease, unadjusted HR: 1.08, 95% CI: 1.06–1.09, *p* < 0.001). After adjusting for confounding variables, patients with higher d-GNRI showed a significantly lower risk of all-cause death, both as a categorical (<92 vs. ≥92, adjusted HR: 1.96, 95% CI: 1.39–2.75, *p* < 0.001) and a continuous variable (adjusted HR: 1.06, 95% CI: 1.04–1.09, *p* < 0.001). On the other hand, a-GNRI was not independently associated with all-cause death (*p* > 0.05).

### 3.4. Predictive Values of a-GNRI and d-GNRI for All-Cause Death

Figure 5 shows the predictive ability of a-GNRI or d-GNRI in terms of risk of all-cause death. The restricted cubic spline curves (Figure 5A) suggest that d-GNRI had a stronger association with the risk of all-cause death both below and over 92 compared to a-GNRI, where higher d-GNRI was more associated with survival than higher a-GNRI, and lower d-GNRI was more associated with death than lower a-GNRI. When comparing the ROC curves of several variables, including a-GNRI, d-GNRI, ∆GNRI, BMI at discharge, ∆BMI, albumin at discharge, and ∆albumin, d-GNRI had the best AUC among those variables, although albumin at discharge also had a high prognostic value close to that of d-GNRI, and there was no statistically significant difference between these two variables by DeLong’s test (*p* = 0.140, Figure 5B). Compared to a-GNRI, d-GNRI had a significantly higher prognostic value for all-cause mortality (*p* < 0.001).

## 4. Discussion

The major findings of the present study are as follows: (1) Although there was a strong correlation between a-GNRI and d-GNRI, GNRI had decreased significantly during the ADHF hospitalization. The changes in GNRI depended more on serum albumin than on BMI. (2) Lower d-GNRI, but not a-GNRI, was independently associated with long-term all-cause mortality in patients with ADHF. (3) Compared to a-GNRI, d-GNRI had more prognostic value for long-term mortality. Together, these findings suggest that the GNRI should be evaluated at the time of hospital discharge rather than at admission among patients hospitalized with ADHF so as to evaluate the patients’ prognosis appropriately.

Nutritional evaluation is of paramount importance for the management of HF patients [21]. Malnutrition is a frequently observed state in patients with acute and chronic HF, and it impacts on patients’ prognosis [22,23]. Nevertheless, there was no consensus or recommendation on how and when to evaluate nutritional status appropriately [24,25]. There are a number of nutritional assessment tools that were associated with prognosis in HF patients. Among them, GNRI seems to have the strongest prognostic impact on the survival of patients with HF [6,26,27,28,29]; lower GNRI was associated with worse prognosis in HF patients. However, those studies evaluated GNRI at the time of the hospital admission mostly [3,4,5,6]. Since both body weight and the serum albumin level are easily influenced by systemic congestion, hemodilution, inflammatory activation, reduced intake, and/or impaired metabolism, GNRI on admission may not necessarily reflect the nutritional status in patients with ADHF [30,31]. In fact, Sze et al. reported that GNRI was the weakest predictor for mortality among several objective nutritional assessment tools, based on the data on admission [32], which was inconsistent with the authors’ other report where GNRI was evaluated among outpatients [26].

Our study demonstrated that d-GNRI had better predictability for all-cause death than a-GNRI (Figure 5 and Table 2). In the present study, despite a strong correlation between a-GNRI and d-GNRI, as expected, there was a substantial decrease in GNRI from admission to discharge, with the absolute mean difference of 2.97. The change was mostly attributed to changes in albumin, while changes in body weight (BMI) had less impact on the GNRI change (Figure 3). In addition, Figure 5B showed that ∆albumin, as well as ∆GNRI, also had a relatively high prognostic value for predicting mortality, whereas ∆BMI, which could partially reflect congestion relief during hospitalization, did not associate with survival. Considering these results, it might be assumed that hydration status by ADHF at hospital admission would not be a key factor for the lower predictive ability of a-GNRI than d-GNRI, and the superiority of d-GNRI over a-GNRI would be attributed more to taking into account the nutritional changes during the critical event of ADHF rather than merely to congestion relief.

Nakayama et al. [33] reported that the mean serum albumin changed from 3.51 mg/dL on day 1 to 3.35 mg/dL on day 7, and the increase was independently associated with favorable outcomes. A similar study was reported by Chao et al. [34] and Gotsman et al. [35], where a decrease in the serum albumin level was associated with mortality in patients with HF. In line with those studies, our study also showed a decrease in the albumin level during hospitalization, although the change was trivial in terms of an average effect (the mean change was only −0.44 mg/dL). In fact, Figure 2 suggested that a certain number of patients experienced an increase in GNRI. Therefore, it is noteworthy that not only the decrease, but also the increase in albumin, as well as GNRI, may play an important role in the reclassification of the nutritional status on an individual basis [10]. In this context, whether or not administration of albumin or other nutritional implements can improve the outcome in patients with lower GNRI may be of great interest, since the efficacy of nutritional interventions on clinical outcomes is still a matter of debate [21]. Our study may pave the way for appropriate measurement of the nutritional status among HF patients and for determining the indication of subsequent nutritional interventions.

As shown in Table 1, patients with lower GNRI (both a-GNRI and d-GNRI) had higher age and more comorbidities, such as hypertension and dyslipidemia, which could be expected since malnutrition is strongly related to the patient’s oldness and comorbidities. Notably, patients with lower d-GNRI had significantly higher LVEF, despite higher BNP/NT-proBNP and a higher prevalence of NYHA class III or IV than those with higher d-GNRI (Table 1). This finding may suggest that malnutrition could be more common in patients with higher LVEF, including heart failure with preserved EF (HFpEF), in which the pathophysiologic mechanisms would include systemic inflammation and may be different from those of HF with reduced EF (HFrEF) [36,37]. In addition, the higher BNP/NT-proBNP or higher prevalence of NYHA class III/IV in the lower d-GNRI group might be attributable to the worse HF condition of this group, exemplified by the worse laboratory data, such as lower hemoglobin and higher CRP, or the longer length of hospital stay. In other words, this result may imply that d-GNRI could reflect the patient’s HF status more precisely than LVEF.

Of noted, the higher prevalence of hemodialysis in the lower GNRI group was observed at discharge, but not at admission (Table 1). Although the cause of the dynamic changes in GNRI during HF hospitalization among hemodialysis patients remains unclear, patients undergoing hemodialysis might be prone to developing malnutrition by an ADHF event. It would be of crucial importance for those patients to assess the nutritional condition at hospital discharge, irrespective of the assessment at hospital admission. Further studies are warranted to elucidate the mechanism of changes in nutritional status during hospitalization and optimal treatments for patients who are at risk of developing malnutrition [9,10].

### Limitations

Our study has several limitations. First, this was a retrospective observational study. Hence, we could not consider unmeasured or unknown variables affecting the results. Second, we could not analyze a lot of patients enrolled in the WET-HF registry due to the lack of data, which might introduce selection bias. Third, we suggest that higher GNRI indicates a better nutritional condition, though we have no additional confirmatory indices of nutritional status from this cohort, such as controlling nutritional status (CONUT) score or prognostic nutritional index (PNI), due to lack of those data. Finally, we do not have data regarding nutritional intervention or the administration of albumin during hospitalization. Past studies have demonstrated the utility of nutritional intervention in HF patients [38]; however, no study has evaluated GNRI as a decision-making tool to determine whether nutritional intervention is needed. Further investigation will be required to elucidate the optimal treatment strategy incorporating nutritional assessment with GNRI in patients presenting with ADHF [39].

## 5. Conclusions

Among patients hospitalized with ADHF, GNRI at the time of hospital discharge was independently associated with long-term all-cause mortality and had more predictability than GNRI at the time of hospital admission. Our study suggested that GNRI should be evaluated at the time of hospital discharge, regardless of the assessment at hospital admission in ADHF patients.

## Figures and Tables

**Figure 1 jcm-12-01891-f001:**
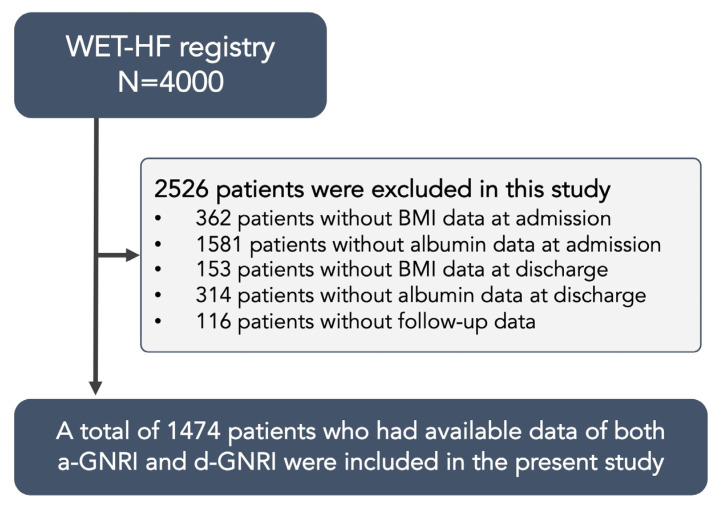
Flowchart of the present study. Patients who were not eligible for the present study were excluded sequentially from the original WET-HF registry.

**Figure 2 jcm-12-01891-f002:**
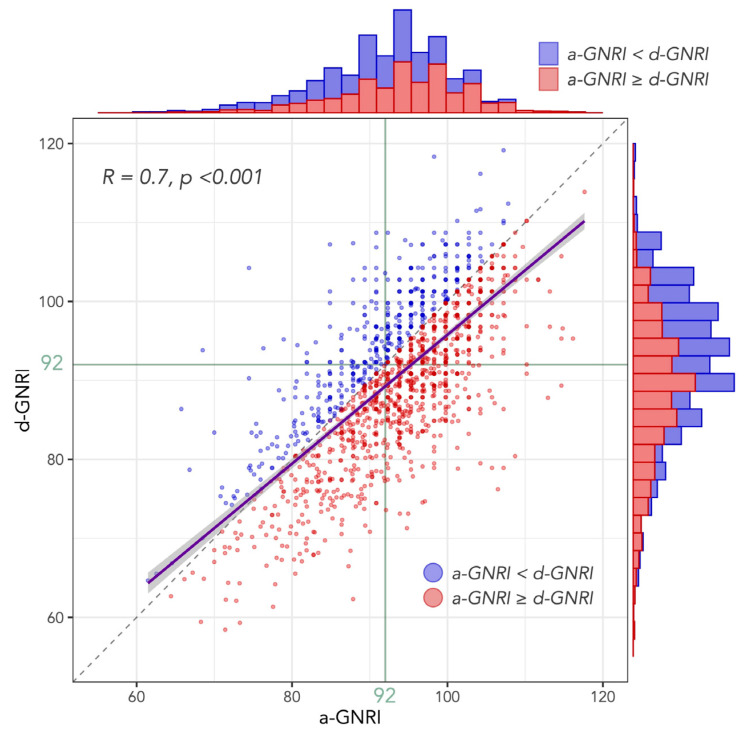
Distribution of GNRI values at hospital admission and discharge. A scatter plot between a-GNRI and d-GNRI with those histograms. Purple line with gray area indicates linear regression line with 95% CI. A blue and red dot indicate a patient with increase and decrease in GNRI during the hospitalization, respectively. While there is strong correlation (R = 0.7) between a-GNRI and d-GNRI, the majority of the patients show decrease in GNRI during the hospitalization.

**Figure 3 jcm-12-01891-f003:**
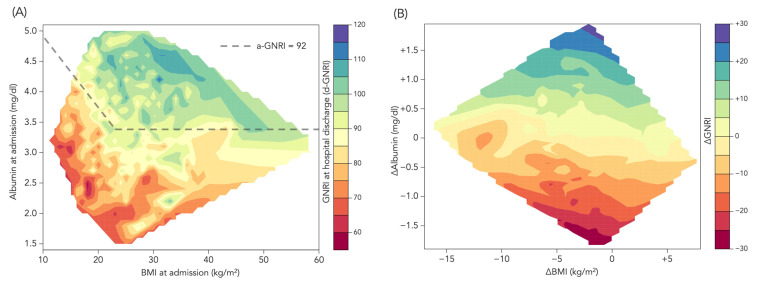
Heatmaps for d-GNRI and ∆GNRI. (**A**) Complementary relationship between BMI and albumin values at hospital admission in predicting the GNRI value at hospital discharge (d-GNRI). d-GNRI is color-coded from red (the lowest d-GNRI) to blue (the highest d-GNRI). The dashed line indicates a-GNRI = 92, so the area above the line indicates a-GNRI > 92, whereas the area below the line indicates a-GNRI < 92. (**B**) ∆GNRI from hospital admission to discharge according to the relationship between ∆BMI and ∆albumin. ∆GNRI is color-coded from red (the lowest decrease) to blue (the highest increase). Because of the formula of GNRI, GNRI completely depends on serum albumin if the patient’s BMI exceeds 22 kg/m^2^ (“body weight/ideal body weight” must be set to 1), and changes in GNRI during hospitalization also depend mainly on albumin rather than BMI.

**Figure 4 jcm-12-01891-f004:**
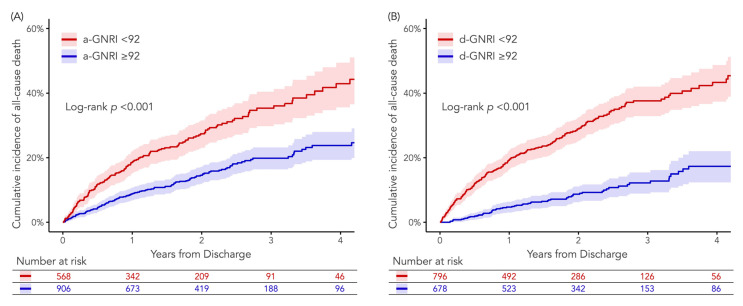
Kaplan–Meier curves for long-term all-cause death according to (**A**) a-GNRI or (**B**) d-GNRI. The red and blue area indicate 95% CIs of the corresponding Kaplan–Meier curves. When stratified by a-GNRI or d-GNRI with the cutoff value of 92, a-GNRI < 92 and d-GNRI < 92 groups had a significantly higher all-cause mortality risk up to 4 years than a-GNRI ≥ 92 and d-GNRI ≥ 92 groups, respectively.

**Figure 5 jcm-12-01891-f005:**
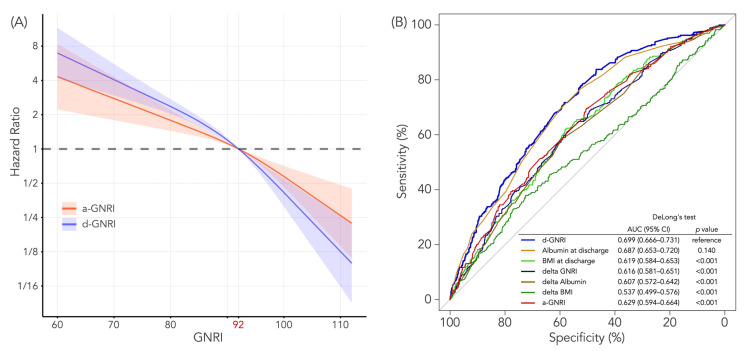
(**A**) Restricted cubic spline curves and (**B**) ROC curves for all-cause death according to a-GNRI or d-GNRI. In Figure (**A**), the reference (i.e., hazard ratio = 1) is the risk on the cutoff value (92) of a-GNRI and d-GNRI. Although the hazard ratios for all-cause death appears to decrease as either a-GNRI or d-GNRI increase, d-GNRI has a more predictive performance for all-cause death than a-GNRI. In Figure (**B**), blue, orange, green, dark blue, brown, dark green, and red curves indicate ROC curves of d-GNRI, albumin at discharge, BMI at discharge, delta GNRI, delta albumin, delta BMI, and a-GNRI, respectively. d-GNRI had the best AUC, which was significantly higher than other variables, except for albumin at discharge, which also had a high prognostic value close to that of d-GNRI.

**Table 1 jcm-12-01891-t001:** Baseline patient characteristics.

Variables	a-GNRI < 92 N = 568	a-GNRI ≥ 92 N = 906	*p*-Value	d-GNRI < 92 N = 796	d-GNRI ≥ 92 N = 678	*p*-Value
GNRI						
at admission	86.4 (81.5–89.3)	98.1 (95.0–101.3)	<0.001	89.6 (83.9–94.0)	98.3 (94.1–102.1)	<0.001
at discharge	83.4 (77.3–89.3)	95.3 (89.8–99.8)	<0.001	84.9 (78.9–88.9)	97.4 (94.7–101.3)	<0.001
Age (years)	79.0 (69.3–85.0)	74.0 (63.0–81.0)	<0.001	79.0 (71.0–85.0)	70.0 (60.0–80.0)	<0.001
Male	51.4 (292/568)	62.9 (570/906)	<0.001	51.4 (409/796)	66.8 (453/678)	<0.001
BMI (kg/m^2^)						
at admission	21.1 (18.5–24.6)	24.0 (21.8–26.6)	<0.001	21.5 (19.1–24.6)	24.7 (22.6–27.5)	<0.001
at discharge	19.3 (17.0–22.2)	22.2 (20.2–24.9)	<0.001	19.7 (17.5–22.3)	22.9 (21.1–25.5)	<0.001
NYHA classification 3 or 4	86.7 (488/563)	79.8 (713/894)	0.001	86.3 (679/787)	77.9 (522/670)	<0.001
Hypertension	72.2 (410/568)	68.7 (622/906)	0.161	73.5 (585/796)	65.9 (447/678)	0.002
Dyslipidemia	38.2 (216/565)	43.9 (395/899)	0.034	40.0 (316/790)	43.8 (295/674)	0.151
Diabetes	36.4 (207/568)	38.6 (350/906)	0.408	38.6 (307/796)	36.9 (250/678)	0.518
Smoking	35.5 (200/563)	42.4 (378/892)	0.010	35.7 (280/785)	44.5 (298/670)	0.001
Hemodialysis	4.0 (23/568)	3.3 (30/905)	0.475	5.5 (44/796)	1.3 (9/677)	<0.001
COPD	5.1 (29/568)	4.4 (40/900)	0.613	5.6 (44/792)	3.7 (25/676)	0.108
History of HF hospitalization	29.8 (169/568)	29.3 (265/905)	0.860	31.5 (251/796)	27.0 (183/677)	0.066
Etiology						
Ischemic	31.0 (176/568)	28.3 (256/906)	0.265	31.0 (247/796)	27.3 (185/678)	0.121
Valvular	25.9 (147/568)	22.2 (201/906)	0.115	27.4 (218/796)	19.2 (130/678)	<0.001
Laboratory data at admission						
Hgb (g/dL)	11.2 (9.8–12.8)	12.5 (10.8–14.2)	<0.001	11.3 (9.9–13.0)	12.8 (11.2–14.3)	<0.001
Na (mEq/L)	139.0 (136.0–142.0)	140.0 (138.0–142.0)	<0.001	139.0 (137.0–142.0)	140.0 (138.0–142.0)	<0.001
BUN (mg/dL)	23.3 (17.1–37.1)	20.7 (16.0–29.0)	<0.001	23.5 (17.6–37.1)	19.9 (15.2–26.7)	<0.001
Alb (mg/dL)	3.2 (2.9–3.4)	3.8 (3.6–4.1)	<0.001	3.4 (3.1–3.7)	3.8 (3.6–4.1)	<0.001
CRP (mg/dL)	1.0 (0.3–3.2)	0.3 (0.1–1.1)	<0.001	0.8 (0.2–2.6)	0.3 (0.1–1.0)	<0.001
BNP (pg/mL)	906.4 (481.6–1602.2)	632.2 (339.3–1060.3)	<0.001	901.0 (481.8–1585.0)	560.3 (314.8–1005.3)	<0.001
NT-proBNP (pg/mL)	5470 (2448–13218)	3203 (1709–6655)	<0.001	5578 (2493–13529)	2736 (1518–5140)	<0.001
Laboratory data at discharge						
Hgb (g/dL)	11.0 (9.8–12.7)	12.4 (10.8–14.0)	<0.001	11.0 (9.8–12.5)	12.9 (11.5–14.5)	<0.001
Na (mEq/L)	139.0 (137.0–141.0)	139.0 (137.0–141.0)	0.008	139.0 (137.0–141.0)	139.0 (137.0–141.0)	0.012
BUN (mg/dL)	23.8 (17.4–35.2)	22.0 (16.5–30.0)	0.001	23.6 (17.0–34.7)	21.6 (16.5–28.7)	<0.001
Alb (mg/dL)	3.2 (2.9–3.5)	3.7 (3.4–4.0)	<0.001	3.2 (2.9–3.4)	3.9 (3.7–4.1)	<0.001
CRP (mg/dL)	0.5 (0.1–1.3)	0.3 (0.1–0.9)	<0.001	0.6 (0.2–1.4)	0.2 (0.1–0.6)	<0.001
BNP (pg/mL)	319.0 (177.2–570.1)	207.2 (102.0–442.3)	<0.001	351.8 (194.9–628.7)	179.0 (94.4–341.0)	<0.001
NT-proBNP (pg/mL)	2932 (1416–7688)	1846 (1022–3205)	0.002	2905 (1609–6585)	1585 (755–2489)	<0.001
LVEF (%)	45.0 (32.0–58.4)	43.9 (31.0–58.0)	0.173	45.8 (32.2–59.0)	40.0 (30.0–57.0)	<0.001
Medications at discharge						
ACEI or ARB	60.7 (345/568)	69.3 (628/906)	0.001	60.4 (481/796)	72.6 (492/678)	<0.001
MRA	38.7 (220/568)	40.9 (370/905)	0.444	37.3 (297/796)	43.3 (293/677)	0.022
beta-blockers	75.7 (429/567)	79.4 (719/906)	0.106	76.6 (609/795)	79.5 (539/678)	0.186
Statines	32.4 (184/568)	36.6 (331/904)	0.104	32.5 (258/795)	38.0 (257/677)	0.028
Length of hospital stay (days)	16 (10–27)	14 (10–21)	<0.001	15 (10–26)	14 (10–21)	0.001

GNRI = geriatric nutritional risk index; BMI = body mass index; NYHA = New York Heart Association; COPD = chronic obstructive pulmonary disease; HF = heart failure; Hgb = hemoglobin; BUN = blood urea nitrogen; Alb = albumin; CRP = C-reactive protein; BNP = brain natriuretic peptide; LVEF = left ventricular ejection fraction; ACEI = angiotensin-converting-enzyme inhibitor; ARB = angiotensin II receptor blocker; MRA = mineralocorticoid receptor antagonists.

**Table 2 jcm-12-01891-t002:** Hazards ratios for all-cause mortality according to Cox regression proportional hazards analysis.

All-Cause Death	Unadjusted HR (95% CI)	*p* Value	Adjusted HR * (95% CI)	*p* Value
a-GNRI				
Categorical (≥92 vs. <92)	2.09 (1.66–2.63)	<0.001	0.77 (0.57–1.05)	0.104
Continuous (per 1 unit decrease)	1.06 (1.04–1.07)	<0.001	0.99 (0.97–1.01)	0.341
d-GNRI				
Categorical (≥92 vs. <92)	3.82 (2.89–5.05)	<0.001	1.96 (1.39–2.75)	<0.001
Continuous (per 1 unit decrease)	1.08 (1.06–1.09)	<0.001	1.06 (1.04–1.09)	<0.001

* Adjusted for age (years), sex, hypertension, dyslipidemia, diabetes mellitus, dialysis, smoking, chronic pulmonary occlusive disease, history of HF admission, New York Heart Association Functional Classification 3 or 4, laboratory data (serum hemoglobin, Na, BUN, serum creatinine) at hospital discharge, left ventricular ejection fraction, prescriptions of beta-blockers, MRA, statin, and either ACE-inhibitor or ARB at hospital discharge, length of index hospitalization, and a-GNRI (for d-GNRI) or d-GNRI (for a-GNRI).

## Data Availability

The data underlying this article cannot be shared publicly to maintain the privacy of the individuals that participated in the study. The data will be shared on reasonable request to the corresponding author.

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
