# Peer review of "Geriatric Nutritional Risk Index at Hospital Admission or Discharge in Patients with Acute Decompensated Heart Failure"

_jcm, 2023, doi:10.3390/jcm12051891_

Round 1
Reviewer 1 Report
1. In this research, the result that a-GNRI was superior to d-GNRI as a prognostic indicator in ADHF patients is considered as novel. However, in order to robustly demonstrate the superiority of GNRI calculated from the value of BMI and albmin as a prognostic indicator, it is necessary to show that indices at discharge other than d-GNRI, especially albumin itself at discharge, BMI itself at discharge, delta-GNRI, or delta-albumin, are inferior to d-GNRI as prognostic indicators.
2. The reason why d-GNRI was a better prognostic indicator than the a-GNRI should be described. It would be important to consider whether the a-GNRI does not accurately reflect nutritional status of the patient in hydration status by ADHF, or whether nutritional status at discharge is important for prognosis rather than that on admission. The former could be examined by using BNP/NT-proBNP-corrected a-GNRI, and the latter by using albumin at discharge as a prognostic indicator.
3. Renal function is a generally recognized indicator related to the prognosis of heart failure. Serum Cr or eGFRneeds to be included in explanatory variables.
4. EF is high in the group with d-GNRI<92, despite higher BNP. This could be included in discussion.
5. In Figure 3 legend, line 4, a-GNRI=22 would be corrected as a-GNRI=92.
Reviewer 2 Report
Dear authors,
to my opinion it is very important to evaluate the association of GNRI with administration of albumin or other nutritional implements during in-hospital stay in order to have explanation why GNRI is more important evaluate at discharge.
Have you considered to add this to your research? Otherwise, how could you explain the value of GNRI which is important to be evaluated just at the discharge.
Round 2
Reviewer 1 Report
The authors have thoughtfully considered the reviewers' comments and significantly improved the manuscript.
The present study clarified that nutritional status at discharge, as assessed by d-GNRI, influences the prognosis of patients hospitalized for acute heart failure, and in particular albumin at discharge has an important impact on prognosis. This result is considered to be a new piece of evidence.